# A Novel Asymmetric Hyperchaotic Image Encryption Scheme Based on Elliptic Curve Cryptography

Haotian Liang , Guidong Zhang *, Wenjin Hou, Pinyi Huang, Bo Liu and Shouliang Li *

School of Information Science and Engineering, Lanzhou University, Lanzhou 730000, China; lianght18@lzu.edu.cn (H.L.); houwj17@lzu.edu.cn (W.H.); huangpy18@lzu.edu.cn (P.H.); liubo17@lzu.edu.cn (B.L.)
* Correspondence: zhanggd@lzu.edu.cn (G.Z.); lishoul@lzu.edu.cn (S.L.); Tel.: +86-185-0948-7799 (S.L.)

**Abstract:** Most of the image encryption schemes based on chaos have so far employed symmetric key cryptography, which leads to a situation where the key cannot be transmitted in public channels, thus limiting their extended application. Based on the elliptic curve cryptography (ECC), we proposed a public key image encryption method where the hash value derived from the plain image was encrypted by ECC. Furthermore, during image permutation, a novel algorithm based on different-sized block was proposed. The plain image was firstly divided into five planes according to the amount of information contained in different bits: the combination of the low 4 bits, and other four planes of high 4 bits respectively. Second, for different planes, the corresponding method of block partition was followed by the rule that the higher the bit plane, the smaller the size of the partitioned block as a basic unit for permutation. In the diffusion phase, the used hyperchaotic sequences in permutation were applied to improve the efficiency. Lots of experimental simulations and cryptanalyses were implemented in which the NPCR and UACI are 99.6124% and 33.4600% respectively, which all suggested that it can effectively resist statistical analysis attacks and chosen plaintext attacks.

**Keywords:** asymmetric encryption; elliptic curve cryptography; different-sized block permutation

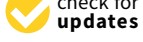

## 1. Introduction

With the sharp growth in network evolution, the communication of multimedia information (i.e., images, audio, and video) over various networks has been effectively facilitated. However, most of the transmission processes for these contents have been done through an unsecure network, which means there is potential risk for the loss of information or the interception (i.e., illegal copying and distribution) or even malicious tampering of this information [1,2]. Therefore, the security of multimedia information, especially in the field of big data and cloud systems, has raised increasing concern [3,4]. Traditional encryption methods include the implementation of the data encryption standard (DES) and advance encryption standard (AES) algorithm. Though the methods mentioned above play a vital role in encryption, the inherent features of the image like the strong correlation of pixels and a large amount of information suggest these preliminary technologies are not suitable. On the basis of the sensitivity to initial situations, ergodicity, mixed characteristics, simple analytic description, and high complex behavior of the chaotic system, quantities of schemes based on hyperchaotic systems have been reported from different researchers.

In 1989, Matthew first proposed a scheme which implemented the chaotic system into the encryption for the text [5]. After Fridrich [6] firstly applied a chaotic system to the digital picture encryption and divided the procedure of encryption into two main parts in 1998, a diversity of algorithms and methods have been proposed based on the two vital process mentioned above: permutation and diffusion. Some of these methods are based on pixel-level permutation, and the nonequilibrium system with chaos was applied into the encryption by constructing some S-boxes [7]. Zhang et al. [8] improved the chaotic map

with delay and cascade. Meanwhile, bit-level permutation has been applied to some other algorithms. In [9], bits of the pixel were segmented into an odd group and an even group, and after that, the chaotic image was calculated by randomly swapping the two groups. A bit-level cyclic shift method was applied in [10] to enhance the effects of the permutation for the plain image. However, Zhou et al. [11] mentioned that when adopting nonlinear transformation directly, due to the narrow interval and unrich nonlinear dynamic behavior, some image encryption algorithms based on low-dimensional chaos systems bear security risks. To overcome this vital shortcoming, the hyperchaotic system was designed and realized for image encryption.

Compared with chaotic system, the hyperchaotic system has stronger randomness that is more suitable for image encryption. In [12], because of the complicated dynamic properties, a four-dimensional (4D) hyperchaotic system was proposed to generate the key stream and control the further rotation and shift for rows and columns of each component of the plain color image. Zhu et al. [13] used a 2-dimensional compound homogeneous hyperchaotic system (CHHCS) in the process of permutation to obtain good effects. However, by using the chosen-plaintext attack, some encrypted images based on hyperchaotic system could still be successfully converted into the corresponding plain image without any error [14,15]. In order to solve this weakness, plaintext was correlated with the chaotic key stream derived from the hyperchaotic sequences when generating the final encryption key stream, which enhanced the security under the chosen-plaintext attack [16]. However, with the improvement of security, the problem of time efficiency has also been exposed.

Over the last several decades, there has been a sharp increase in the interest in promoting the time efficiency of hyperchaotic image encryption. By using the chaotic shift transform (CST), Liu et al. [17] combined the processes of permutation and diffusion into one stage, which effectively enhanced the time efficiency. As an important part of the attempt to improve the time efficiency, block shuffling has raised much more attention in recent studies. The author of [18] presented the algorithm where the image was divided into 4 blocks, and after the replacement according to the pseudorandom sequences derived from the combination chaotic system, these subblocks are combined together. Another scheme proposed in [19] has divided each color constituent into the same sized blocks with the application of the substitution based on the paired S-box.

Moreover, the consideration of permutation on the bit-level instead of the perspectives of pixel-level has also grown rapidly. To reduce the redundancies implied by the Fridrich's structure, a novel circular inter- and intrapixel bit-level permutation strategy was mentioned in [20]. Aqeel-ur-Rehman et al. [21] proposed a 2-bit-level pixel substitution by using DNA and chaos. However, there are vital flaws for the methods mentioned above: (1) Although the use of block surely reduced the time redundancy, all these referred block sizes are not determined by the property of the image. (2) All the bits of each pixel were treated equally without the consideration of the discrepancy of the quantity of information they respectively contained.

Nevertheless, though these proposed schemes proved to be of high security and efficiency in the process of image encryption, there is one thing they certainly have in common: all these schemes are symmetric. This significant attribute results in one inevitable flaw: the important and sensitive key must be transmitted through a dedicated secure channel to prevent eavesdropping. That means if the key is disclosed, security for further communication is impossible [22].

In view of the shortcomings above, we propose a novel asymmetric hyperchaotic image encryption scheme based on elliptic curve cryptography (ECC). The author of [23] illustrated the good time efficiency and security of ECC theoretically, and Rajdeep et al. [24] made an analysis of various encryption systems, in which ECC shows its excellent characteristics. The advantages of this scheme are as follows: (1) Based on the high security of ECC, sensitive data related to the initial parameters can be transmitted with the cipher image through the public channels. (2) With the application of block-level bit permutation, after the permutation process, because the permutated planes are bit-level planes, the place

of the pixels in the image and the values of each pixel are both therefore changed, which means better time efficiency of the encryption with the even better effects. (3) During the diffusion process, the same hyperchaotic sequences used in permutation significantly reduces the complexity of the algorithm and provides better storage efficiency.

The remainder of the paper is arranged as follows. In Section 2, elliptic curve integrate encrypt scheme (ECIES) for the encryption of the sensitive data is applied and the hyperchaotic system is briefly introduced. In Section 3, we described this novel asymmetric hyperchaotic image encryption scheme based on ECC in detail, followed by the experimental simulation in Section 4. Finally, the conclusions are presented in Section 5.

## 2. Preliminary Knowledge

### 2.1. ECEIS Method

With the fast development of the public key system, Victor Miller [25] and Neal Koblitz [26] independently proposed a cryptosystem based on elliptic curves in 1985. Subsequently, Menezes et al. [27] discussed the elliptic curve cryptosystems and their implementation, which leads to the Menezes–Vanstone cryptosystem. Compared with other public key encryption systems like RSA, ECC presented better performance in terms of efficiency and security due to the significantly shorter keys. Despite these good attributes being shown, however, in [28], under certain conditions, the Menezes–Vanstone cryptosystem is insecure, which means it could not be considered a probabilistic encryption algorithm. On the basis of the Diffie–Hellman augmented encryption scheme (DHAES) presented in [29] in 1998, ECIES was presented in [30], which contained the vital attributes of the DHAES with the usage of elliptic curves in an integrated scheme: (1) public key operations, (2) symmetric encryption algorithms, (3) MAC codes, and (4) hash computations.

Martinez et al. discussed a variety of the ECIES, and introduced the functions of the basic ECIES:

- Key agreement (KA): Function used for the generation of a shared secret by two parties.
- Key derivation (KDF): Mechanism that produces a set of keys from keying material and some optional parameters.
- Hash (HASH): Digest function.
- Encryption (ENC): Symmetric encryption algorithm.
- Message authentication code (MAC): Information used to authenticate a message [31].

The whole encryption procedure was narrated below:

1. Before starting the process, a pair of temporary keys must be generated by the sender. We assume that the private key in this key pair will be named as u, and the name for the public key will be U.
2. By using the sender's temporary private key and the recipient's public key (we define it as V here and the private key is v) together, a shared secret value is obtained through the key agreement function, KA.
3. The shared secret value (and optionally with other parameters) will be taken as the input data for the KDF (key derivation function). After the process of the KDF, the encryption key, noted as $k_{ENC}$, and the MAC key, noted as $k_{MAC}$, are both obtained in the format of the concatenation.
4. By using the ENC symmetric algorithm and the encryption key $k_{ENC}$, the plain message will be encrypted as a cipher message (Here we denoted as c).
5. With the fabrication through the MAC function, a tag will be produced by the input of the MAC key and the encrypted message c.
6. Finally, the sender will concatenate these three parts together: (1) the temporary public key U; (2) the encrypted message c; and (3) the tag. The final message which will be sent to the recipient is in the following format: U||c||tag [32].

For the recipient, by using the reversed procedure, the plain message will be obtained.

The specific flow charts for encryption and decryption of the ECIES are shown in Figures 1 and 2.

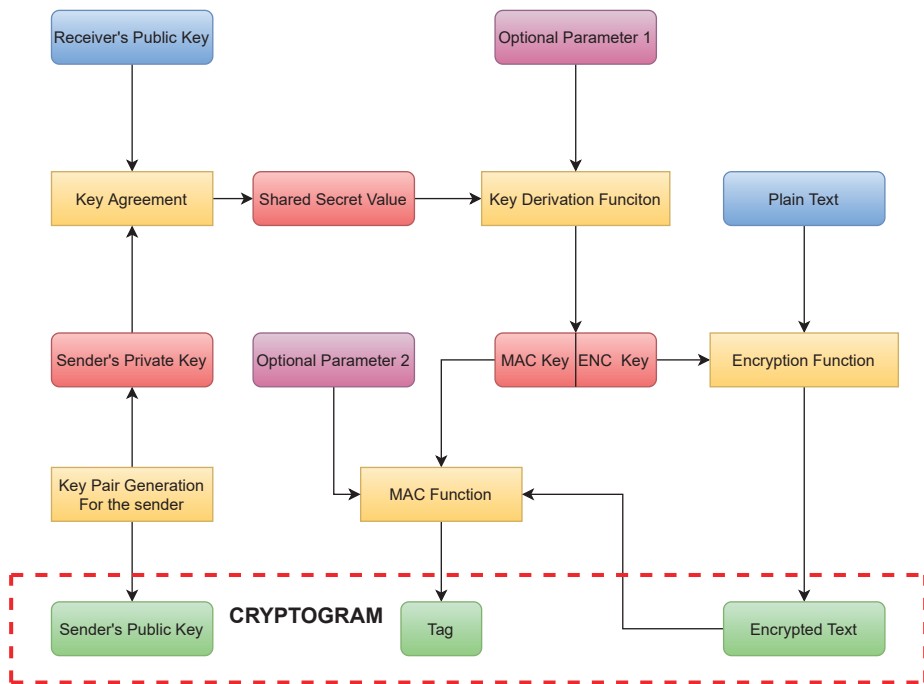

**Figure 1.** The encryption process of ECIES.

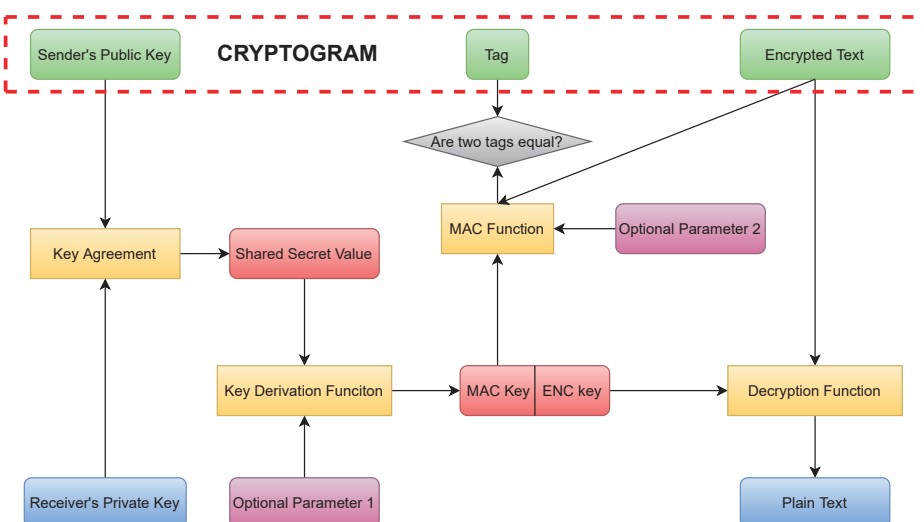

**Figure 2.** The decryption process of ECIES.

In this paper, an open-source python application provided by ecies.org is applied to the encryption of the sensitive data related with the initial parameters of the hyperchaotic system.

### 2.2. Hyperchaotic System

For the encryption schemes based on chaotic systems, the corresponding chaotic systems play an important role in generating the pseudorandom sequences for permutation and diffusion. Due to the better pseudo-randomness and higher unpredictability, the logistic Feigenbaum non-linear cross-coupled hyperchaotic map (LF-NCHM) proposed by the authors of [33] is used here. The mathematical expression is shown in (1), where the parameter $\mu \in (0, 4]$.

$$\begin{cases} x_n = mod(\mu \times (\delta \times y_{n-1}) \times (1 - \delta \times y_{n-1}), 1) \\ y_n = mod(\lambda \times sin(\pi \times (x_{n-1} + y_{n-1})), 1) \end{cases} \tag{1}$$



### 3. The Asymmetric Hyperchaotic Image Encryption Scheme

In this scheme, we suppose that the sender wants to transmit a confidential image through the public channel to the receiver, and the whole procedure is summarized as follow. First, the sender encrypts the plaintext image using hash value derived from the original image, and the cipher image is obtained through the permutation and diffusion. Then, the received public key from the receiver is used to encrypt the hash value by ECIES, followed by the process of combining it with encrypted images into encrypted packet. After that, the encrypted packet is transferred to the receiver through the public channel. When the receiver gets the encrypted data packet, the decrypted hash value is obtained by the use of the private key and ECIES. Finally, the receiver can use the hash value and the decryption scheme to extract the corresponding plaintext image. The whole process of the scheme is shown in Figure 3.

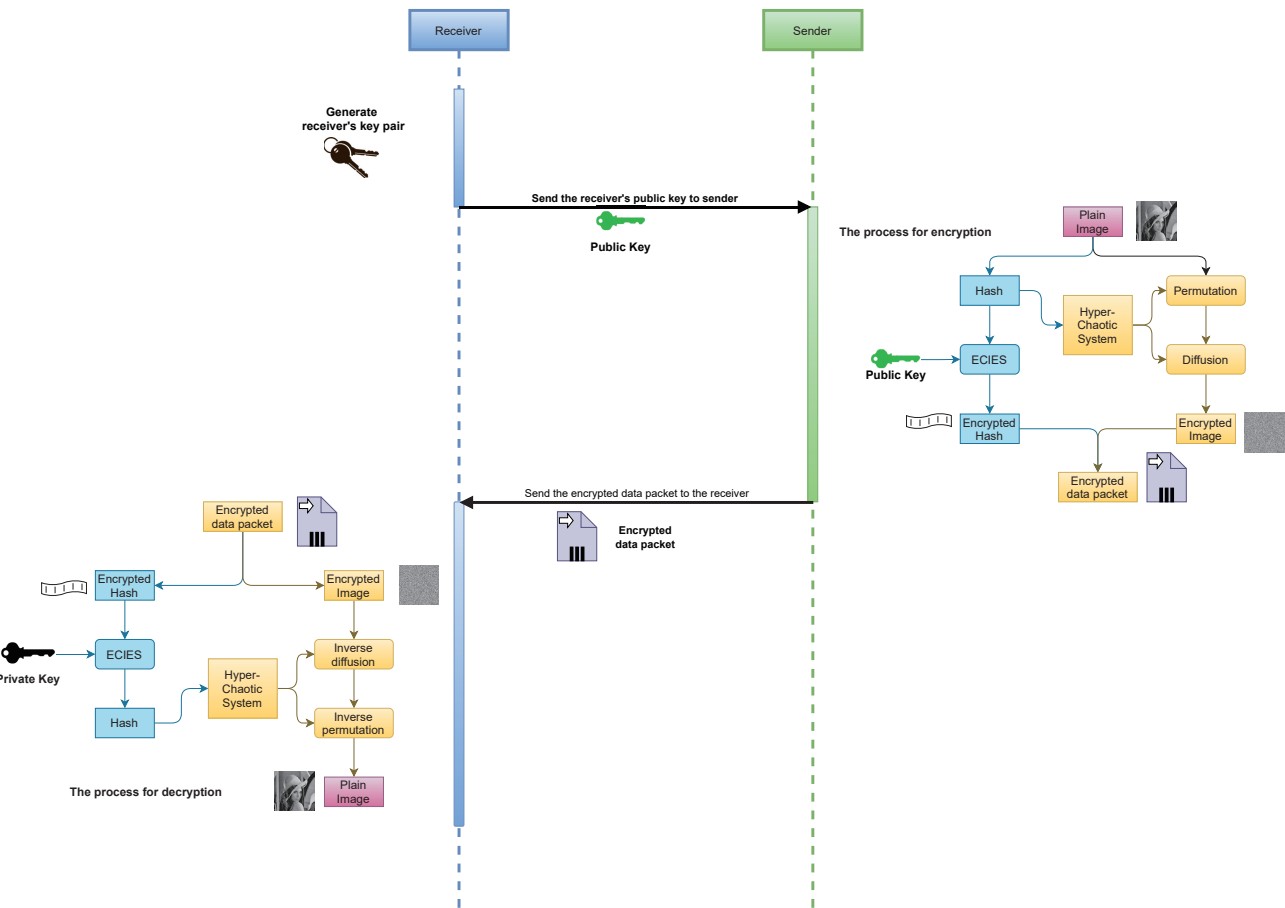

**Figure 3.** An example for the whole transmission process.

As for the encryption algorithm, it can be generally divided into two components: the permutation and the diffusion. For the permutation step, not only has the place of each pixel in the image changed, but with the separation and the combination of the bits in each pixel, the value of each pixel has also changed, which demonstrates the additional effects of diffusion. Moreover, with the consideration of the amount of information that the different bits in the same pixel contain, the corresponding divided bit-level planes have also been parted into the different sized block units for the following step of permutating, making both time efficiency and the strength of security strongly enhanced.

In the process of diffusion, for the advanced utilization ratio of the chaotic sequences, the same sequences applied in permutation are reused. In a two-dimensional image, the method proposed in [31] is applied to augment the complexity of the correlation

between different pixels. It is started from the bottom to the top and from the right to the left, and after a round of diffusion, for each pixel, it is related with two proximal pixels in the downward and rightward direction.

The specific procedure of the algorithm is displayed in Figure 4, and the part in the green frame is the novel different-sized blocks permutation proposed in this paper.

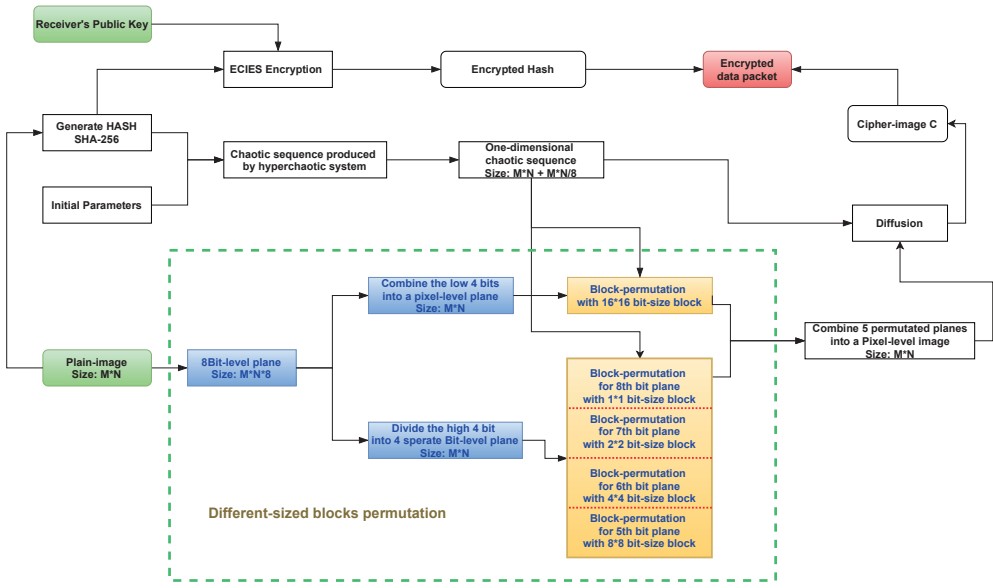

**Figure 4.** The general flow chart of the algorithm.

### 3.1. The Formation of the Key and the Hyperchaotic Sequences

Before the generation of the hyperchaotic sequences by using the LF-NCHM, the initial values were generated based on the hash value derived from the plaintext image. To ensure security, we used the SHA-256 algorithm to obtain the dynamic hash which is related to the plaintext image, which indicates that even the most trivial alteration would still result in a huge difference for the result. On the basis of the dynamic hash, the initial conditions with extremely strong sensitivity were then determined. After that, these conditions were treated as the input of the LF-NCHM for the construction of the hyperchaotic sequences used in further operations. The specific steps are as follows:

1. Generate the hash value by using the plaintext image, then split the obtained hash into 32 pieces, like $K_1, K_2 \ldots K_{32}$, and each of these pieces contains 8 bits of data related to the plaintext image.
2. With the application of the formulas shown in (2), calculate the intermediate values $h_1, h_2, h_3, h_4$, and the inputs of the formula are $K_1, K_2 \ldots K_{32}$ from Step 1.

$$
\begin{cases}
h_1 = \dfrac{(k_{25} \oplus k_{26} \oplus \ldots \oplus k_{32})}{256} \\
h_2 = \dfrac{(k_1 \oplus k_2 \oplus \ldots \oplus k_8)}{256} \\
h_3 = \dfrac{(k_{17} \oplus k_{18} \oplus \ldots \oplus k_{24})}{256} \\
h_4 = \dfrac{(k_9 \oplus k_{10} \oplus \ldots \oplus k_{16})}{256}
\end{cases}
\tag{2}
$$

3. Calculate the initial conditions $\mu$, $\lambda$, $x_0$, and $y_0$ in accordance with the $h_1, h_2, h_3, h_4,$ and the equations applied are displayed in (3).

$$\begin{cases} \mu = mod(\mu_0' + \dfrac{(h_1 + h_2)}{256}, 4) \\[2mm] \lambda = mod(\lambda_0' + \dfrac{(h_2 + h_3)}{256}, 6) \\[2mm] x_0 = mod(x_0' + \dfrac{(h_3 + h_4) \times 10^{14}}{256}, 1) \\[2mm] y_0 = mod(y_0' + \dfrac{(h_4 + h_1) \times 10^{14}}{256}, 1) \end{cases} \tag{3}$$

In the formula, the $\mu_0'$, $\lambda_0'$, $x_0'$, and $y_0'$ are the initial values given in advance. For the algorithm in this scheme, these conditions are designated here: $\mu_0' = 3.1$, $\lambda_0' = 4.4$, $x_0' = 0.3$, and $y_0' = 0.5$.

4. Take the obtained initial values as the input of LF-NCHM to generate the hyperchaotic sequences with the length needed.

### 3.2. The Permutation Process Based on Different Sized Blocks

In the permutation process, the plaintext image P with the size $M \times N$ will firstly be converted into 5 planes on bit-level from the original image with the pixel-level format. The contents of the plane set are a "combined" plane in which all components are the combination of the 1st–4th bit of the pixels in the original image and four planes representing the 5th–8th bit of the pixels, respectively. After converting the image for each content in the set, it would be parted into smaller blocks when it contained more information compared with other planes. More specifically, in this chapter, the plane which is composed of the 8th bit for all the pixels will be divided into $1 \times 1$ blocks, and for the 7th, 6th, and 5th bit, the corresponding plane will be divided with the size of $2 \times 2$, $4 \times 4$, and $8 \times 8$, respectively. For the "combined" plane in the set, because the four bits in this plane contain less information than others in the set, the block size will be $16 \times 16$ to improve the operating speed. Next, the planes are shuffled like the matrixes based on the unit of the blocks divided before, and the distance for shuffling is mainly determined by the hyperchaotic sequences. Finally, combine these 5 permutated planes together and convert into the pixel-level image. Through this novel different sized block permutation, the image experienced both permutation and diffusion simultaneously.

The concrete steps are as follows:

1. Convert the original plaintext image into 5 different planes, the first plane is the combination of the 1st–4th bits of each pixel in the plaintext image, and the latter 4 planes are made of the corresponding 5th–8th bits respectively. One example for the regulation referred to before is presented in Figure 5.

2. Divide these planes into the different sized blocks according to the amount of information that each bit included. In this chapter, the plane of the 8th bit of the plane which contains more information than any other bits in every pixel, so the size of block is $1 \times 1$. Based on this consideration, when the length and width of the planes are both the multiples of 16, the 7th, 6th, and 5th plane are divided into the different blocks with the size of $2 \times 2$, $4 \times 4$, and $8 \times 8$, respectively. Otherwise, the planes will be added by zeros. Finally, for the "combined" plane mentioned before, due to it containing the least amount of information, $16 \times 16$ is the proper block size in order to enhance the time efficiency. The specific process of the division for an example applied with the $2 \times 2$ size is displayed in Figure 6.

3. To ensure the randomness and the relation with the plaintext image, the hyperchaotic sequences generated before are used to determine the distance of the shuffling op-

eration. Firstly, to avoid the insecurity of the over repeated usage of the sequences, the formula (4) for calculating the shifting displacement is used.

$$\begin{cases} \text{row} = floor(mod(rowChao(indexRow^2 + n) \times 10^4, indexCol)) \\ \text{col} = floor(mod(colChao(indexCol^2 + n) \times 10^4, indexRow)) \end{cases} \quad (4)$$

where the indexRow and the indexCol are the numbers of the rows and the columns for the amount of the blocks. The rowChao and colChao are two obtained hyperchaotic sequences, the n means the index of the present column or the row, and the floor(x) obtains the largest integer which is less than or equal to x.

After the value of the migration distance is determined, the shuffling process is as follows: all the rows of the matrix after division will shift in turn, first according to the distance calculated in advance, then the columns will also shift in accordance with their migration distance analogously. An example to interpret the process of shuffling based on the matrix in Figure 6 is shown in Figure 7.

4. Combine all 5 planes after permutation into one pixel-level image PM with the size $M \times N$, which is essentially the inverted process of Step 1 essentially. When it is done, the operation of permutation is finished.

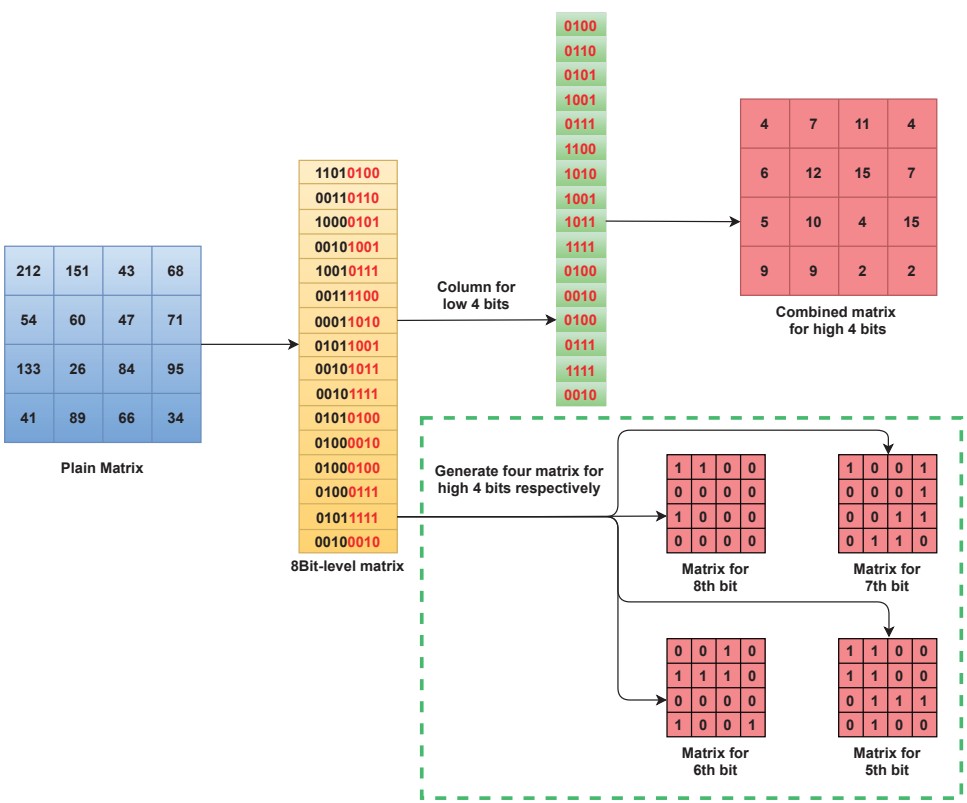

**Figure 5.** The conversion of the plaintext image.

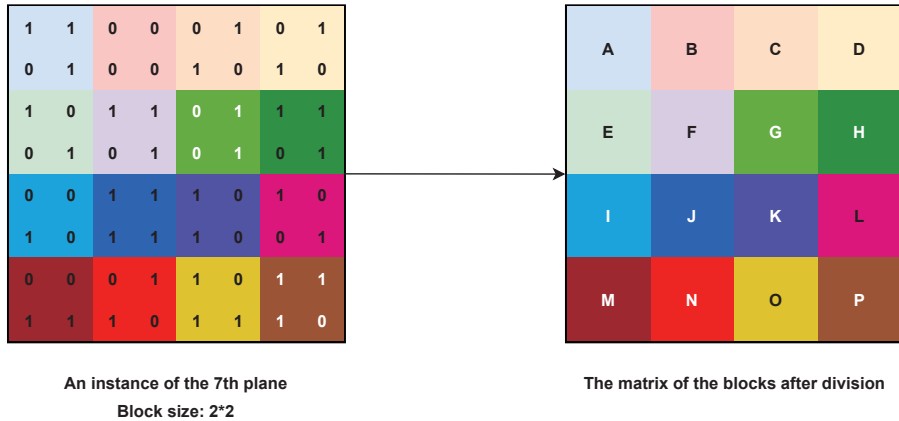

**Figure 6.** An instance of division for the plane where the blocks after division are named from A to P.

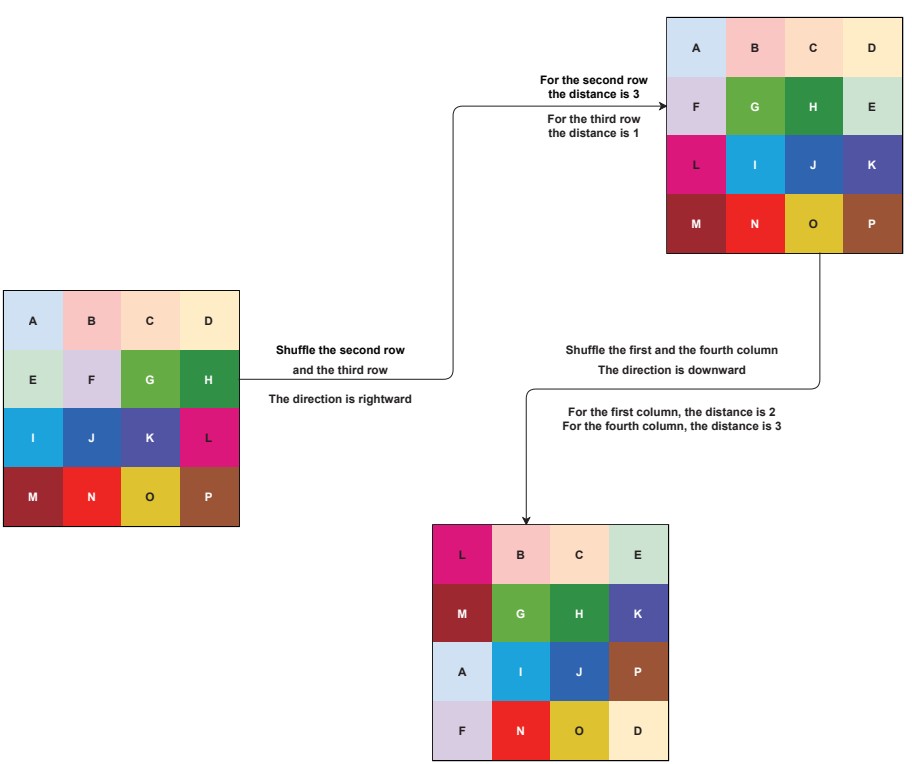

**Figure 7.** The example of the shuffling process based on Figure 6.

### 3.3. The Diffusion Process

To reduce the correlation between the neighboring pixels in the image, it is necessary to take the diffusion process. For the diffusion process, there are two major categories: block diffusion and the one based on the pixel-by-pixel mode. For the algorithm proposed in this article, we adopt a one-round operation of the pixel-by-pixel diffusion. The operation is started from the pixel located in the lower right corner of the two-dimensional image, end up with the pixel of the upper left corner of the image. In the whole operation, the spread process is from the botom to the top, and from the right to the left, which means after the present pixel is relevant to its lower and right neighbors. The details of the operation based on [33] are as follows:

1. With the generation of the hyperchaotic sequences, the new subsequence for the usage of the diffusion process will be intercepted from one of the obtained sequences. In this

article, the chosen sequence for interception is x(i), and a subsequence was intercepted from the x(i). After that, the subsequence x(i) was reshaped into a two-dimensional sequence x(i, j) with the size $M \times N$.

2. As the input of the diffusion process, the permutated image PM(i, j) was treated through this subprocess: for each pixel in the permutated image, the relation with the right and lower adjacent pixels was built. The final product after the diffusion is the cipher image C(i, j). The specific algorithm is shown in Algorithm 1.

---

**Algorithm 1** The process of diffusion

---

1: **for** $i = M; i \geq 1; i - -$ **do**
2:     **for** $j = N; j \geq 1; j - -$ **do**
3:         **if** $i == M \&\& j == N$ **then**
4:             C(i,j) = PM(i,j)$\oplus$floor(x(i,j)$\times$255)
5:         **else if** $i == M$ **then**
6:             $diffu_1$ = PM(i,j)$\oplus$C(i,j+1)
7:             C(i,j) = $diffu_1 \oplus$floor(x(i,j)$\times$255)
8:         **else if** $j == N$ **then**
9:             $diffu_1$ = PM(i,j)$\oplus$C(i+1,j)
10:             C(i,j) = $diffu_1 \oplus$floor(x(i,j)$\times$255)
11:         **else**
12:             $diffu_1$ = PM(i,j)$\oplus$C(i,j+1)
13:             $diffu_2$ = $diffu_1 \oplus$C(i+1,j)
14:             C(i,j) = $diffu_2 \oplus$floor(x(i,j)$\times$255)
15:         **end if**
16:     **end for**
17: **end for**

---

In this part, an example with four pixels was shown to provide the convenience of the comprehension for the method elaborated above. Figure 8 shows the steps of this operation:

$$C(2,2) = PM(2,2) \bigoplus floor(x(2,2) \times 255)) = 110$$
$$C(2,1) = PM(2,1) \bigoplus C(2,2) \bigoplus floor(x(2,1) \times 255)) = 59$$
$$C(1,2) = PM(1,2) \bigoplus C(2,2) \bigoplus floor(x(1,2) \times 255)) = 11$$
$$C(1,1) = PM(1,1) \bigoplus C(1,2) \bigoplus C(2,1) \bigoplus floor(x(1,1) \times 255)) = 114$$

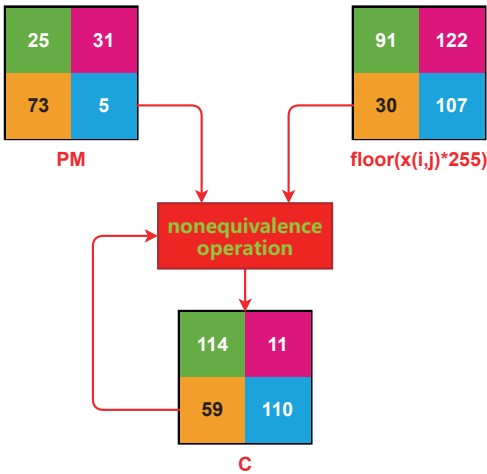

**Figure 8.** An example of the diffusion.

### 3.4. The Formation of the Encrypted Data Packet

1.  After the encryption process for the plaintext image, the hash value that generates the hyperchaotic system was encrypted through the ECIES, with the public key received from the receiver. The encrypted hash and image were combined together to form an encrypted data packet (PKT) for the final transmission. The presence of the PKT signifies the end of the whole encryption operation.

### 3.5. Decryption Process

In general, the decryption process of the scheme proposed is basically the reverse of the encryption. The input for the decryption process is the receiver's private key u and the PKT.

1.  Split the PKT into the encrypted hash and the encrypted image.
2.  Use the private key u to decrypt the encrypted hash.
3.  Use the decrypted hash from the last step to generate the hyperchaotic sequences x(i) and y(i), and reshape the x(i) into the two-dimensional sequence x(i, j).
4.  Take the inverse diffusion process to obtained the permutated-only image PM from the cipher image C and the use of the sequence x(i, j). The concrete process is as follows:
5.  With the combination of two sequences x(i) and y(i), take the reversed operation of the permutation process to obtain five planes which are the divided parts of the original image.
6.  Finally, build them all up into one thing and restore it to the decrypted image P, which size is $M \times N$ as the same of the original plaintext image.

Figure 9 reveals the flow chart of the decryption procedure.

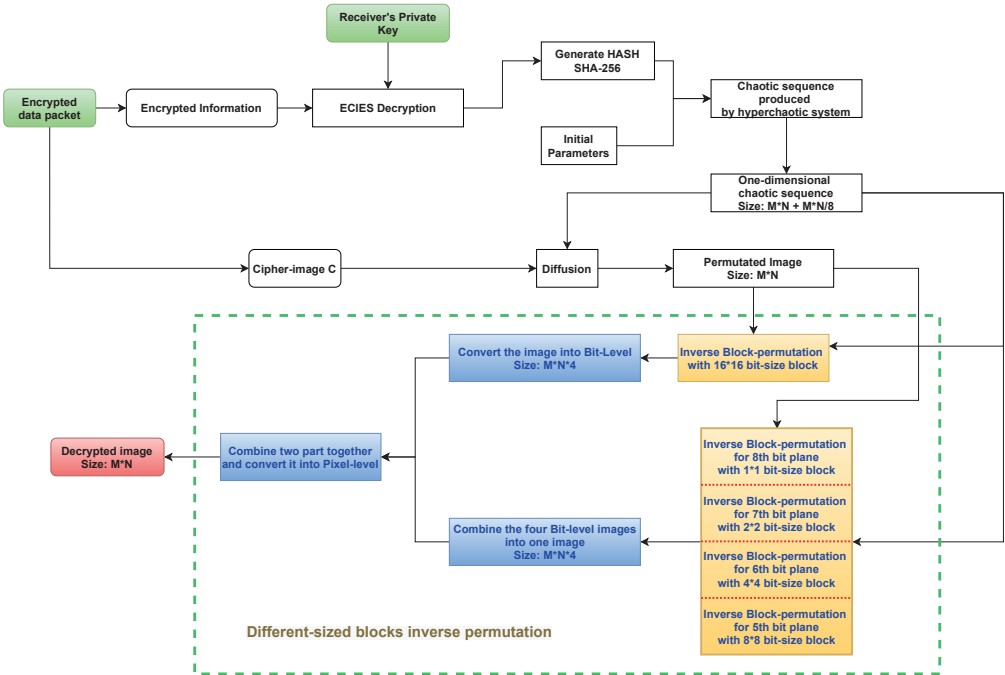

**Figure 9.** The concrete flow chart for the decryption process.

## 4. Simulation Results and Attack Test

### 4.1. Simulation Results

In this part, the experimental environment is listed here: the software platforms are MATLAB R2016b and PyCharm 2020.3.2; the hardware is Inter (R) Core (TM) i7-8650 CPU @ 2.11 GHz with 8 GB RAM; and the operating system is Windows 10. Four grayscale images with a size of $256 \times 256$ are used: Lena, Cameraman, Pepper, and Baboon. The simulation results are shown in Figure 10, (a) is the original image of Lena, (b) and (c)

correspond to the encrypted and decrypted image of Lena. The three groups of pictures are displayed below as (d–f), (g–i), (j–l) respectively. It is obvious that the image obtained after the encryption shows it randomness, and it is impossible to understand the original meaning through human vision, which remarkably demonstrates the excellent effect of the encryption. Moreover, by using the decryption algorithm, the encrypted image can be converted into the corresponding original plaintext image with great efficiency. From the experimental result, the original secret information can be precisely restored without any difference and loss, which illustrates the applicability and validity of the whole process.

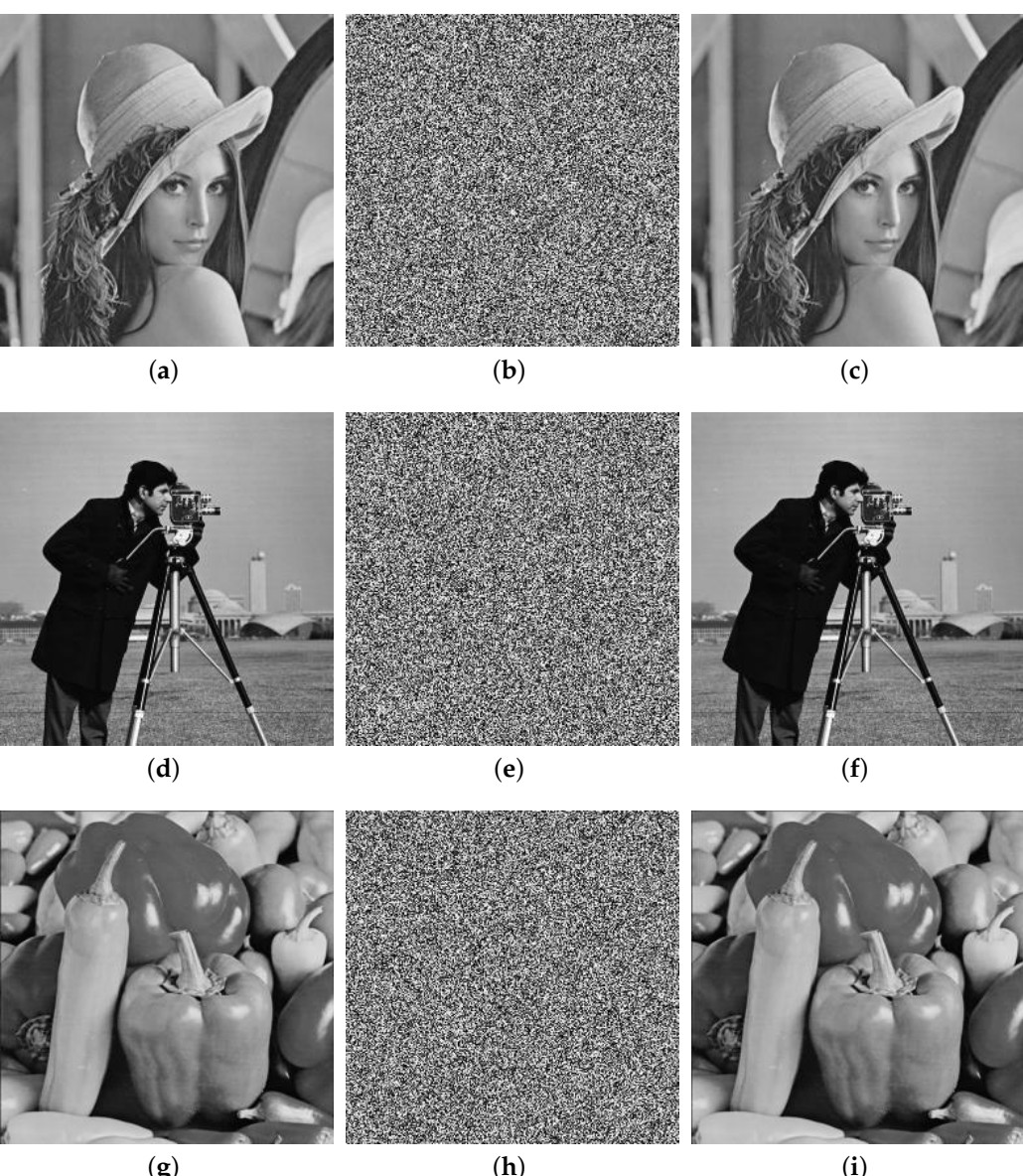

**Figure 10.** *Cont.*

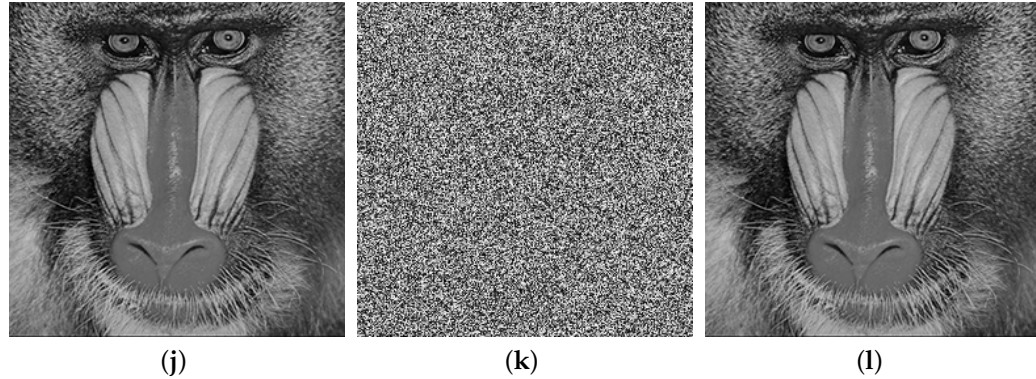

(j)　　　　　　　　　　　　(k)　　　　　　　　　　　　(l)

**Figure 10.** The simulation test for the images. (**a**–**c**) are the original image, encrypted image and decrypted image of the Lena image, respectively. (**d**–**f**) are the original image, encrypted image and decrypted image of the Cameraman image, respectively. (**g**–**i**) are the original image, encrypted image and decrypted image of the Pepper image, respectively. (**j**–**l**) are the original image, encrypted image and decrypted image of the Baboon image, respectively.

*4.2. Statistical Analysis*

4.2.1. Histogram Analysis

　　The distribution of gray values from a digital image can be effectively and visually revealed through the histogram of the image. In order to satisfy the aim of the encryption for an image, the histogram of the encrypted image should almost be or close uniform and, nevertheless, it should show its prominent difference from the one derived from the plaintext image, because when distribution of the gray value is more even, it will be harder for the attacker to obtain information from the encrypted image through statistical attacks. The histogram of the four images mentioned above and their encrypted versions are shown in Figure 11. The obvious characteristics of the plaintext image are clearly displayed through the histogram. Meanwhile, the distribution of the histogram derived from the cipher image is reasonably uniform, which will precisely reduce the correlation of the adjacent pixels and there will be no information provided for the attackers.

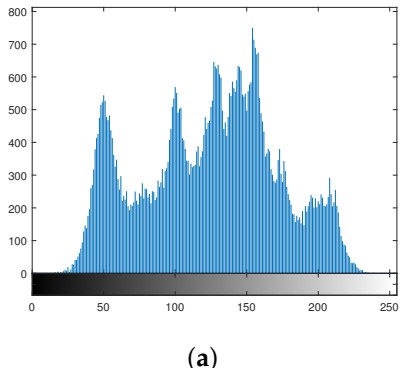

(a)

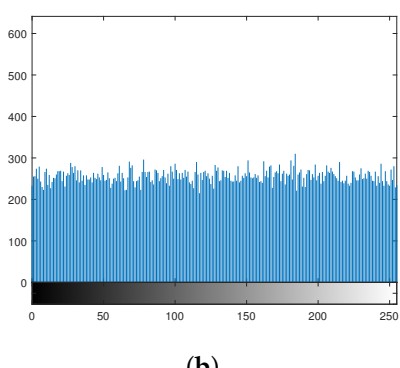

(b)

**Figure 11.** *Cont.*

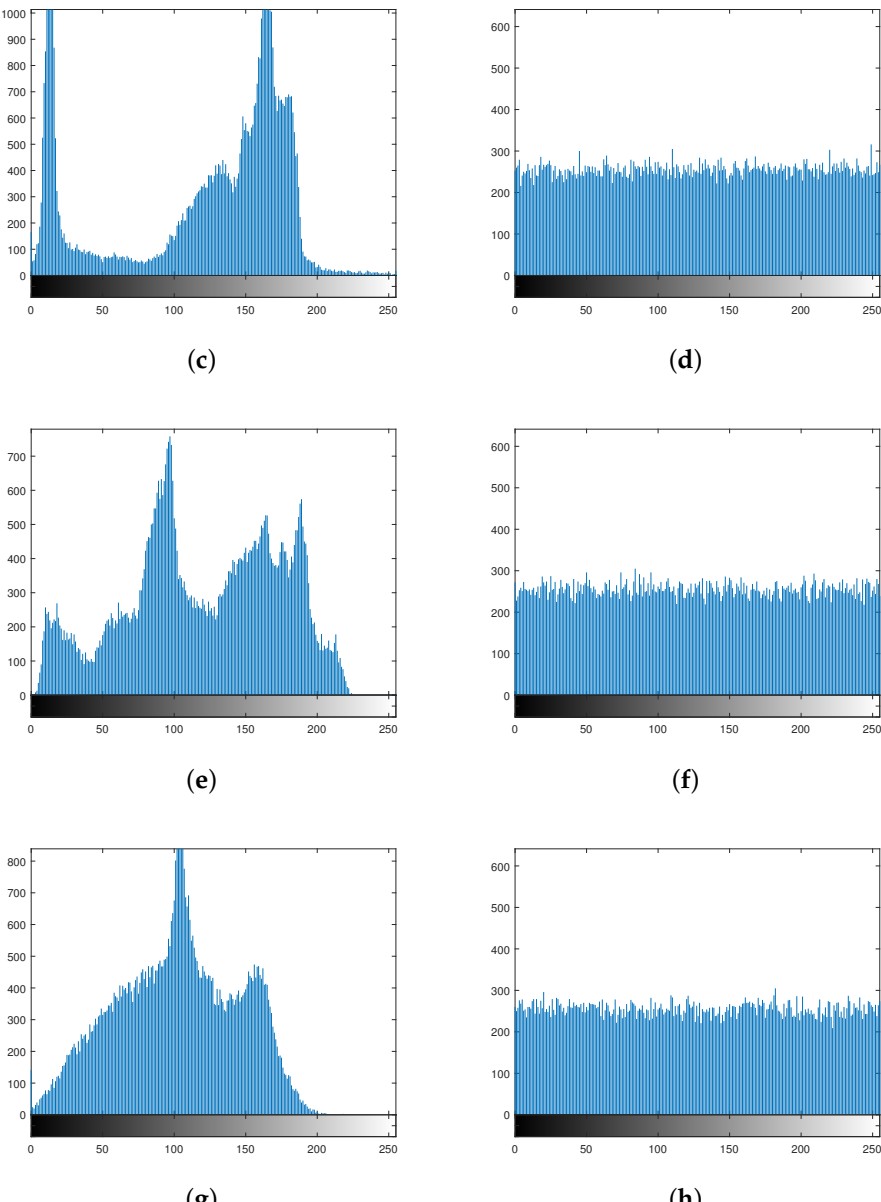

**Figure 11.** Histograms. (**a**,**b**) are histograms of the Lena image and the encrypted Lena image. (**c**,**d**) are histograms of the Cameraman image and the encrypted Cameraman image. (**e**,**f**) are histograms of the Pepper image and the encrypted Pepper image. (**g**,**h**) are histograms of the Baboon image and the encrypted Baboon image.

### 4.2.2. Correlation Analysis

In general, the coefficient correlation between two adjacent different pixels in a variety of plaintext images are quite strong, which means it is necessary to eliminate such correlation by using an efficient and secure encryption algorithm. The aim of a low coefficient correlation among the neighboring pixels should be achieved in the corresponding cipher image after the encryption process for the plaintext image. The formula for calculating the correlation between two different contiguous pixels is displayed below as (5). We choose the pairs of pixels in the plaintext image and the cipher image in three different directions: horizontal, vertical, and diagonal. The coefficient correlation of the plaintext image and the corresponding cipher image in three directions were obtained by using Formula (5). The test results of the correlation in three directions for the plaintext images and the images after encryption are shown in Table 1. The consequences of the correlation coefficients

of the comparison among the schemes in [20,34,35] are shown in Table 2. From the displayed data, we know the results of the correlation are close to zero in the cipher image, which means the correlations are significantly reduced. Moreover, it is obvious that our encryption scheme performed better compared with the other schemes mentioned before.

$$p_{xy} = \frac{E[x - E(x)][y - E(y)]}{D(x)D(y)} \tag{5}$$

The distribution of the contiguous pixels for the plaintext image and the encrypted image are revealed in Figure 12, to avouch the randomness of the simulation test, we stochastically select nearly 10,000 pairs of pixels from the image in three directions. Through the analysis of the figure which shows the distribution, it is clear that the values of the adjacent pixels in all three directions are completely randomized after the encryption. This illustrates the reliable defensiveness under the statistical attack, which means for the attackers, their attempts to gain statistical information from the cipher images are futile. In conclusion, our scheme performed well through the statistical analysis.

**Table 1.** Correlation coefficients of different images.

|  | Horizontal | Vertical | Diagonal |
|---|---|---|---|
| Lena | 0.9449 | 0.9728 | 0.9210 |
| Encrypted Lena | −0.0023 | −0.0017 | −0.0022 |
| Cameraman | 0.9330 | 0.9592 | 0.9074 |
| Encrypted Cameraman | −0.0023 | 0.0035 | −0.0053 |
| Pepper | 0.9633 | 0.9706 | 0.9360 |
| Encrypted Pepper | 0.0044 | −0.0008 | 0.0019 |
| Baboon | 0.7985 | 0.7419 | 0.6893 |
| Encrypted Baboon | −0.0042 | 0.0010 | −0.0010 |

**Table 2.** Correlation coefficients of the Lena images encrypted with diverse schemes.

|  | Horizontal | Vertical | Diagonal |
|---|---|---|---|
| Lena | 0.9449 | 0.9728 | 0.9210 |
| Our Encrypted Lena | **−0.0023** | **−0.0017** | **−0.0022** |
| [20] | 0.0064 | 0.0035 | 0.0035 |
| [34] | 0.0034 | −0.0076 | 0.0062 |
| [35] | 0.0024 | −0.0086 | 0.0402 |

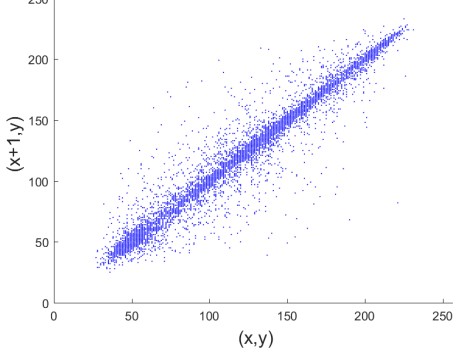 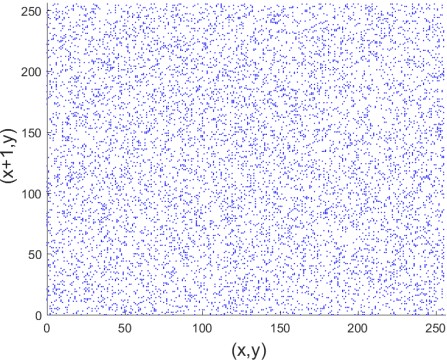

**Figure 12.** *Cont.*

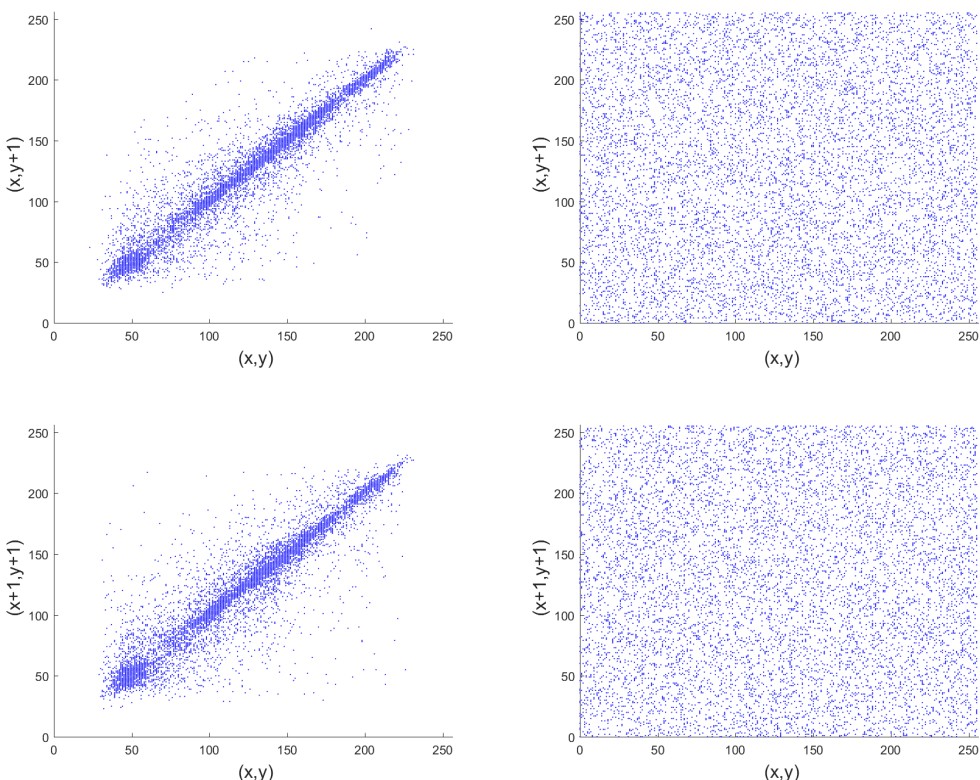

**Figure 12.** Distributions of the pixels in the Lena image.

### 4.3. Key Space Analysis

Broadly speaking, for the encryption system, with the growth of the key space, the security of the algorithm will be relatively strengthened . In this scheme, the core part is the hyperchaotic system, whose attribute is vitally decided through the initial parameters, and the provided initial values $\mu_0$, $\lambda_0$, $x_0$, and $y_0$. Moreover, these values are derived through the hash value, which is generated from the plaintext image and securely transmitted by using the ECIES for encryption. The key space of spec256k1 (https://en.bitcoin.it/wiki/Secp256k1, accessed on 31 March 2021), Advanced Encryption Standard-256-Galois/Counter Mode (AEC-256-GCM) in the ECIES, and the hash value from the plaintext image are all $2^{256}$. With the consideration of the computing power of current devices, it is impossible for attackers to use either public channel attack or brute force attack to steal the key from users. As a result, the analysis demonstrates that the key space of this novel scheme is secure enough.

### 4.4. Sensitivity Analysis

#### 4.4.1. Key Sensitivity

Generally speaking, the key sensitivity is a vital feature for the security of an image encryption scheme because it demonstrates the intensity for resisting brute force attack. According to the generation method of the key, when there is even a little change in the plaintext image, the corresponding image after the encryption is completely different. Nevertheless, a slight variation in the hash value will also result in two diverse cipher images, and the plaintext cannot be extracted from the cipher image when using a modified key whose attribute has been slightly changed. Due to the high sensitivity to the input parameters, which is a remarkable property of the hyperchaotic system, even the slightest variation will result in the situation mentioned above.

The original image of Lena, the corresponding image after encryption, the image after decryption with the correct key, and the three images by using the hash value with only 1 bit changed for the input of the decryption process are display in Figure 13. These

false decryption results demonstrate that even if there is a small change in the hash value, the image cannot be correctly decrypted. As a result, this novel scheme has strong key sensitivity.

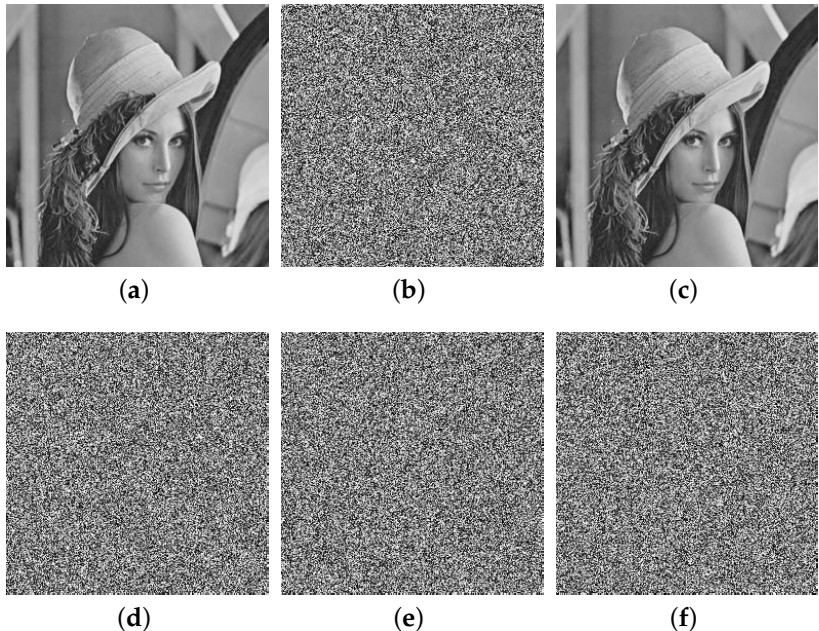

**Figure 13.** Testing results for key sensitivity. (**a**) Lena, (**b**) the encrypted image of Lena, (**c**) the decrypted image of Lena, (**d**–**f**) three examples of the decryption result when a random bit of the hash is changed randomly.

4.4.2. Differential Attack

For many attackers, in order to find the breakpoint of the encryption scheme, they might change a pixel of the plaintext image slightly and analyze the difference between the original cipher image and the newly encrypted one. To prevent such situations that the attackers can find some interesting relations among the cipher image and the encrypted modified images, it is necessary for an image encryption algorithm to perform a sensitive reaction even if there only one single bit has changed. Various methods for testing this type of attribute has been proposed, and in these methods, the number of pixel change rate (NPCR), unified average changing intensity (UACI) [36], and the block average changing intensity (BACI) are applied. The theoretical values of these evaluating indicators are 99.6094%, 33.4635%, and 26.7712%, respectively.

For the calculating procession of these three indicators mentioned above, we first normally encrypted the plaintext image, the cipher image after encryption would be named C. After that, we just change a facultative pixel of the plaintext image and execute the encryption operation again to acquire the cipher image $C'$. Finally, these two images will be the input of the formulas shown in (6)–(8) to get the experimental results of the three indicators. The whole process will be executed 15 times, where the changed pixels are randomly different each time. After 15 rounds of process, the average results for three criteria of the four images are shown in Table 3. Compared with the theoretical values, our experimental results illustrate the excellent strength in resisting the differential attack.

$$\text{NPCR}(P_1, P_2) = \frac{1}{MN} \sum_{i=1}^{M} \sum_{j=1}^{N} |Sign(P_1(i,j) - P_2(i,j))| \times 100\% \tag{6}$$

$$\text{UACI}(P_1, P_2) = \frac{1}{MN} \sum_{i=1}^{M} \sum_{j=1}^{N} \frac{|P_1(i,j) - P_2(i,j)|}{255 - 0} \times 100\% \tag{7}$$

$$\text{BACI}(\text{P}_1, \text{P}_2) = \frac{1}{(M-1)(N-1)} \sum_{i=1}^{(M-1)(N-1)} \frac{m_i}{255} \times 100\% \qquad (8)$$

In these formulas, $M \times N$ is the size of the image, Sign(x) is the symbol function, and the $m_i$ shows the average of the absolute values of the difference between the two elements.

Moreover, Table 4 displays the results of the number of pixel change rate (NPCR) and unified average changing intensity (UACI) values in this paper and other references. It is evident to see that from the comparison result, the NPCR results performs better than [20,34,37,38], and inferior to [1,33], but for the UACI results, the value in our algorithm is closer to the theoretical value than for any other algorithms. Considering all these three implications, the algorithm proposed here performs well in resisting the differential attacks.

**Table 3.** The analytical results for the plaintext sensitivity.

| Index | Lena | Cameraman | Pepper | Baboon | Theoretical Value |
|-------|------|-----------|--------|--------|-------------------|
| NPCR | 99.6124 | 99.6118 | 99.6124 | 99.6031 | 99.6094 |
| UACI | 33.4600 | 33.4697 | 33.4618 | 33.4589 | 33.4635 |
| BACI | 26.7639 | 26.7681 | 26.7619 | 26.7674 | 26.7712 |

**Table 4.** The comparison between the different algorithms for the plaintext sensitivity.

| | NPCR | UACI |
|---|------|------|
| Ours | **99.6124** | **33.4600** |
| [1] | 99.6109 | 33.4354 |
| [20] | 99.5705 | 33.4781 |
| [33] | 99.6078 | 33.4404 |
| [34] | 99.6013 | 33.6531 |
| [37] | 99.6262 | 33.4384 |
| [38] | 99.6192 | 33.5316 |

*4.5. Information Entropy Analysis*

One of the important features to show the randomness of the image and measure the encryption scheme is the information entropy, which reveals the confusion degree of the image. The equation that calculated the information entropy is displayed below:

$$\text{H}(\text{m}) = -\sum_{i=0}^{L} p(m_i) \log_2 p(m_i) \qquad (9)$$

where $p(m)$ represents the probability of symbol $m$, which is the source of the information. H(m) shows the value of the entropy, and $L$ is the amount of grayscale pixels of the image. The theoretical value of information entropy H is 8.

With the growth in the entropy of the image, much more uncertainty is also shown. The closer it approaches to the ideal value 8, the harder it is for the attackers to divulge information from the image. Table 5 reveals the enthalpy values of the four images and their corresponding encrypted images. From the values in the table, we can see that after the encryption, the entropy for all four cipher images is quite close to the ideal theoretical value and is entirely different from the values in the relevant plaintext image, which means that the algorithm proposed here has significantly disordered the pixels in the digital image.

*4.6. Security of Transmission on Public Channels*

From many hyperchaotic encryption schemes presented in recent years, although various boosts have been applied to improve the security of the scheme, for both sides of the communication, vital and sensitive data related to the generation of the hyperchaotic system and other confidential things must be transmitted through the dedicated

secure channel or agreed in advance to avoid the potential risk of the eavesdropping from the malicious participants on this communication channel. In order to increase the convenience for the transmission, according to the scheme proposed in this article, sensitive hash data will be encrypted through the ECIES before transmission through the public channel. With their combination, strong defensiveness is provided by spec256k1 (https://en.bitcoin.it/wiki/Secp256k1, accessed on 31 March 2021) and the advanced encryption standard-256-Galois/counter mode (AES-256-GCM), which have been extensively used due to their soundness in resisting a variety of different attacks. Moreover, the usage of the one-time key pair and the MAC function which offers validation for the tag in the cipher message also strengthens the security of the ECIES system. In conclusion, it is obvious that this encryption scheme shows improvements in the security of transmission on public channels.

**Table 5.** The information entropy of the variant images.

|  | Entropy |
| --- | --- |
| Lena | 7.4486 |
| Encrypted Lena | 7.9971 |
| Cameraman | 7.1047 |
| Encrypted Cameraman | 7.9974 |
| Pepper | 7.5936 |
| Encrypted Pepper | 7.9969 |
| Baboon | 7.3739 |
| Encrypted Baboon | 7.9972 |

*4.7. Time Efficiency of the Encryption Scheme*

In addition to security, time efficiency is also an important indicator for an encryption scheme. A practical encryption should strike a balance between the aspects of security effect and time efficiency, which indicates that it is expected to have both time and effect advantages. Many different criteria have been proposed in recent years; however, nearly all these time-evaluating methods show their drawbacks in diverse situations, and to make our judgment of the time efficiency more comprehensive, the encryption time, encryption throughput (ET), and number of cycles were applied into the assessment of the time efficiency of our algorithm. ET is negatively correlated with the complexity of the algorithm, while the number of cycles shows the positive correlation with the computational complexity. The calculation formula for the ET and the formula for the number of cycles is displayed below:

$$\text{ET} = \frac{image_{size}(byte)}{encryption_{time}(second)} \tag{10}$$

$$\text{Number of cycles per byte} = \frac{CPU_{speed}(Hertz)}{ET(byte)} \tag{11}$$

In this paper, the Lena image with size $256 \times 256$ is adopted for the test case, and 100 times encryption is executed for calculating the average time of our algorithm.

Table 6 shows the encryption time of our algorithm and some other algorithms in [33,37,38], and the comparison of the EF and the number of cycles is revealed in Table 7. From the table we can see that the algorithm we proposed performs with better time efficiency and consumes fewer machine cycles. Compared with the block-based scheme in [38], our scheme improved the security with the exchange of the speed, while the time efficiency is enhanced from the comparison with the pixel-level algorithm in [33] and the bit-level algorithms in [21,37]. Considering the results from the comparison, our algorithm performed excellently in both time efficiency and security aspects.

**Table 6.** The encryption time for different algorithms (seconds).

|  | 256 × 256 | Platform |
|---|---|---|
| Ours | 0.4313 | MATLAB |
| [21] | 0.5612 | MATLAB |
| [33] | 0.7389 | MATLAB |
| [37] | 0.5358 | MATLAB |
| [38] | 0.3065 | MATLAB |

**Table 7.** A comparison of the efficiency of different algorithms.

|  | Encryption Throughput (MBps) | Cycles per Byte |
|---|---|---|
| Ours | **0.2029** | **10,394.31** |
| [21] | 0.1560 | 13,525.64 |
| [33] | 0.1184 | 17,807.46 |
| [37] | 0.1634 | 12,912.76 |
| [38] | 0.2856 | 7386.63 |

## 5. Conclusions

In this paper, a novel asymmetric hyperchaotic image encryption scheme based on elliptic curve cryptography was proposed. To avoid public channel attacks, in accordance with good speed and security, ECC was used to encrypt sensitive data related with the initial parameters which generated the hyperchaotic sequence.

For the enhancement of time efficiency, in permutation the plain image was firstly divided into five planes according to the amount of information contained in different bits: the combination of the low 4 bits, and the other four planes of high 4 bits. After that, the corresponding rules of block partition were adopted for different planes: smaller sized blocks for the plane contained more information than others. This was followed by the permutation in accordance with the hyperchaotic sequences, where these partied blocks were regarded as the basic units. In the process of diffusion, for the improvement of simplicity of the algorithm, the hyperchaotic sequence used in permutation was applied in diffusion. For the pixels in the two-dimensional plane, we started from the bottom right corner, right to left, bottom to top, and after this procedure, pixels were generally related to the pixels in the previous two directions, which results in more complexity for the correlation.

The experimental results demonstrated that this scheme could resist public channel attacks, statistical analysis attacks, and chosen-plaintext attacks with advantages. According to the data from the analysis of encryption speed (throughput and the number of machine cycles consumed), the scheme performed better speed compared with some bit-level based scheme and showed more security than a various of block shuffling methods. The current version of the algorithm is implemented on the basis of the MATLAB platform, and to improve the time efficiency, the FPGA version will be considered soon for further progress.

**Author Contributions:** During the completion of this article, S.L. and W.H. guided the determination of the direction for the research and the revising of the whole article; G.Z. offered the opportunity and the guidance of the knowledge that related with the hyperchaotic encryption; B.L. provided some advice and ideas for the implementation on the MATLAB during the experimental simulation; H.L. constructed the structure of the scheme, performed the experiments with the data mining and completed the whole paper; P.H. contributed to the composition with the use of texLive. All authors have read and agreed to the published version of the manuscript.

**Funding:** This research was funded by the Fundamental Research Funds for the Central Universities (lzujbky-2018-126).

**Acknowledgments:** In the whole process of accomplishing this paper, I really appreciate the support from my family and the help from colleagues of Lanzhou University, who also contributed a lot. Finally, I also sincerely thank the foundation for the research support.

**Conflicts of Interest:** The authors declare no conflicts of interest.

## Abbreviations

The following abbreviations are used in this manuscript:

| **Abbreviations** | |
|---|---|
| ECC | elliptic curve cryptography |
| DES | data encryption standard |
| AES | advance encryption standard |
| CHHCS | compound homogeneous hyperchaotic system |
| CST | chaotic shift transform |
| ECIES | elliptic curve integrate encrypt scheme |
| DHAES | Diffie–Hellman augmented encryption scheme |
| LF-NCHM | logistic Feigenbaum non-linear cross-coupled hyperchaotic map |
| NPCR | number of pixel change rate |
| UACI | unified average changing intensity |
| BACI | block average changing intensity |
| AES-256-GCM | advanced encryption standard-256-Galois/counter mode |
| ET | encryption throughput |
| **Symbols** | |
| $k_{ENC}$ | The encryption key in ECIES |
| $k_{MAC}$ | The MAC key in ECIES |
| u | The sender's private key in ECIES |
| U | The sender's public key in ECIES |
| v | The receiver's private key in ECIES |
| V | The receiver's public key in ECIES |
| c | The intermediate encrypted message in ECIES |
| tag | The production from the MAC function in ECIES |
| $x_0, y_0, \mu, \lambda, \delta$ | The parameters in LF-NCHM |
| $\mu, \lambda, x_0, y_0$ | The initial conditions for the hyperchaotic system |
| $\mu_0', \lambda_0', x_0', y_0'$ | The initial values given in advance for calculating the corresponding conditions |
| M, N | The length and the width of the image in simulation experiment |
| rowChao, colChao | The two obtained hyperchaotic sequences |
| indexRow, indexCol | The number of the rows and the columns for the amount of the blocks |
| row, col | The shifting distance for the rows and the columns |
| PM | The image after permutation |
| C | The cipher image after the permutation and diffusion |
| PKT | The formed encryption data packet after whole process |
| spec256k1 | The parameters of the elliptic curve used in Bitcoin's public-key cryptography |
| $C'$ | The image after the value of a random pixel in the cipher iamge has changed |

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
