# Peer review of "A Novel Asymmetric Hyperchaotic Image Encryption Scheme Based on Elliptic Curve Cryptography"

_applsci, doi:10.3390/app11125691_

Round 1
Reviewer 1 Report
The paper proposes asymmetric hyperchaotic image encryption scheme based on elliptic curve cryptography. The research methodology is clear. However, the paper can be improved in the following ways:
1- A table of abbreviations used.
2- Also, a table of symbols used throughout the paper can help the reader in finding certain symbols and what they mean.
3- The flow charts for encryption and decryption of the previously proposed scheme ECIES on page 4 should be centered the middle of the page.
4- Figure 3 should be centered in the middle of the page.
5- It will help the reader to see the simulation environment used in the study stated before the results and attack test.
Reviewer 2 Report
The manuscript is about hyperchaotic image encryption using elliptic curve cryptography. It is well written. On the other hands, the authors should improve the manuscript.
Basic reporting
The work is somehow new, but several limitations from the technical hinder to grasp the main point of the work, some of them are listed below:
+ In my opinion, the abstract is too cumbersome, and It is hard to catch the key point.
+ In the abstract, authors must report the numerical values finding from the results sections.
+ It is essential to address their method using the algorithm, which makes it clear to grasp the steps of the improvements of the technique.
Experimental design
+ The time and space complexity and algorithm not specified.
+ Test Setup and tuning for the work are expected to elaborate and detailed for future productions.
+ You need to add one Table to show all algorithm's parameters that tuned in this research.
+ There is no flowchart. The authors need to add a detailed flow chart to show all steps
The validity of the findings
+ What is your justification for using your method?
Comments for the Author
+ Overall good work, well done.
+ The literature has to be strongly updated with some relevant and recent papers focused on the fields dealt with the manuscript [1].
[1] CPP-ELM: cryptographically privacy-preserving extreme learning machine for cloud systems. International Journal of Computational Intelligence Systems, 11(1), 33-44.
Round 2
Reviewer 2 Report
The authors answered all questions. In my opinion, the manuscript is acceptable for the journal.